# Understanding reticence to occupy free, novel-design homes: A qualitative study in Mtwara, Southeast Tanzania

Judith Meta[1,2], Salum Mshamu[2,3]*, Salma Halifa[2], Arnold Mmbando[4,5], Hannah Sloan Wood[6], Otis Sloan Wood[6], Thomas Chevalier Bøjstrup[6], Nicholas P. J. Day[3,7], Jakob Knudsen[6], Steven W. Lindsay[5], Jacqueline Deen[8], Lorenz von Seidlein[3,7], Christopher Pell[9,10,11]

1 Independent Consultant and Social Scientist, Mtwara, Tanzania, 2 CSK Research Solutions, Mtwara, Tanzania, 3 Nuffield Department of Clinical Medicine, University of Oxford, Oxford, United Kingdom, 4 Ifakara Health Institute, Ifakara, Tanzania, 5 Department of Biosciences, Durham University, Durham, United Kingdom, 6 Royal Danish Academy, Architecture, Design, Conservation, Copenhagen, Denmark, 7 Mahidol-Oxford Tropical Medicine Research Unit (MORU), Bangkok, Thailand, 8 University of Philippines, Manila, Philippines, 9 Amsterdam Institute for Global Health and Development (AIGHD), Amsterdam, the Netherlands, 10 Department of Global Health, Amsterdam UMC, location University of Amsterdam, Amsterdam, the Netherlands, 11 Amsterdam Public Health Research Institute (Global Health | Health Behaviours & Chronic Diseases Research), Amsterdam, the Netherlands

* salum.mshamu@gmail.com, smshamu@csk.co.tz, salum.mshamu@ox.ac.uk

## Abstract

### Introduction

The population of Africa set to reach 2 billion by 2050. There is therefore great demand for housing across the continent. Research on modified novel designs for housing is a priority to ensure that these homes are not sites of infection for diseases transmission such as malaria. One trial to assess the protection afforded by novel design houses is underway in Mtwara Region, southeastern Tanzania. After constructing 110 of such homes across 60 villages, project staff encountered a certain reticence of the target population to occupy the homes and were faced with accusations of having nefarious intentions. This article explores these accusations, their impacts on home occupancy and lessons for future housing studies.

### Methods

This qualitative study drew on in-depth interviews and focus group discussions with ten occupants of the intervention homes, six community leaders and a further 24 community members. Interviews were recorded, transcribed verbatim and translated to English for qualitative content analysis.

### Results

In communities around the Star Homes, during construction and handover, project staff were widely associated with 'Freemasons', a term used to practices, secrecy, and other conspiracy theories in rural Tanzania. These connections were attributed to other community

**Data Availability Statement:** All data presented in this article will be accessible whenever required.

Requests can be made through National Research Ethics Committee of Tanzania. Data is not available openly for ethical reasons, particularly because of the risk of identifying respondents and sensitive nature of some of the information. The National Research Ethics Committee can be accessed through National Institute for Medical Research, 3 Barack Obama Drive, Box 9653 11101 Dar es salaam. Email: ethics@nimr.or.tz. Project reference number NIMR/HQ/R.8c/Vol.I/2383.

**Funding:** This study is funded by Hanako Foundation, Singapore and sponsored by University of Oxford. The prime grant holder is University of Oxford. The funders had no role in study design, data collection and analysis, decision to publish, or preparation of the manuscript.

**Competing interests:** The authors have declared that no competing interests exist.

members and explained in terms of knowledge deficit or envy, with others hoping to be allocated the home. The stories were embedded in assumptions of reciprocity and suspicions about study motives, linked to limited experience of research. The relationship between the accusations of freemasonry and reticence to occupy the houses was not straightforward, with project staff or relatives playing a role in decisions. The stakes were high, because the recipients of Star Homes were the poorest families in targeted communities.

## Conclusion

The results indicate the need for long-term and proactive community engagement, which focuses on building relationships and providing information through recognizable voices and formats. Given the stakes at play in housing interventions, research teams should be prepared for the social upheaval the provision of free new housing can cause.

## Background

The population of sub-Saharan Africa is growing more rapidly than any other region and is expected to rise from 1.3 billion today to 2 billion in 2050 [1]. Demand for new housing will therefore escalate in cities, particularly smaller cities, where growth is greatest, and in rural populations, albeit at a slower pace [2]. The home meanwhile remains a common location for the transmission of infections suffered by children across Africa: malaria, diarrhoea, and respiratory tract infections. Mosquitoes transmit malaria largely while children sleep indoors at night; unhygienic toilets predispose to enteric infections; and polluted air from indoor cooking increases the risk of respiratory tract infections. Better housing construction can reduce these health risks and, considering the population size at risk, even small health improvements could make a large impact [3, 4].

To explore which housing improvements are feasible and beneficial in rural Africa, a multidisciplinary research team carried out two sets of studies of potential designs. Between 2014 and 2015 a pilot study of six novel designs were constructed in Magoda village, north-east Tanzania [5]. Magoda was chosen based on an experience on research and developmental projects over more than two decades, including the construction of a village dispensary by one of the project collaborators (JK). Guided by south-east Asian rural house designs [6], the buildings were elevated off the ground, either as single-storey buildings on stilts or as double-storey buildings. Two of the houses, one single and one double-storey building were each clad in air permeable materials. All study houses had improved airflow and reduced mosquito densities compared to traditional housing [5].

This experience illustrated that fundamental Asian building principles of good ventilation and elevated bedrooms, were feasible in a rural African village, such as Magoda. In-depth interviews and focus group discussions with residents in new and traditional homes in Magoda confirmed the overall acceptance and informed design priorities for the next generation of house designs [5]. Biannual visits over the following five years showed that the new houses were utilized by residents and no substantial reconfigurations had been undertaken. The success of the pilot led to an interdisciplinary collaboration between architects, clinicians, entomologists, public health advocates and social scientists to develop and evaluate novel housing designs for rural Africa [4, 7].

High grade evidence–ideally from randomized controlled trials (RCTs), comparing intervention and control houses–is needed to demonstrate that novel house designs lead to

improved health. Based on the pilot study and taking into account the preferences of residents, the team designed a novel healthy house and set out to run a RCT in rural Tanzania. The new 'Star Home' was designed to protect against three diseases of childhood, responsibility for great childhood mortality in sub-Saharan Africa: malaria, diarrhoea and acute respiratory tract infections [8, 9]. Using data from the national malaria control programme and after fact-finding visits, Mtwara Region in south-east Tanzania was selected as a study site: in contrast to the pilot study site in Tanga, malaria transmission remained high and there were few other ongoing and planned health-related studies.

Despite funding and logistical challenges, by early 2021, the construction of 110 Star Homes had been completed across 60 villages in Mtwara Region. With strict eligibility criteria that targeted the poorest families in the communities, households had been selected based on a lottery system [9]. Keys were handed over to the lottery winners (who had been consulted throughout construction) and the study staff were ready to begin assessing the health impact of these new homes. However, around 10% of new homeowners did not move in and many more used their new houses inconsistently. Researchers were confronted with stories of "Freemasons" and accused of nefarious motives. The research team set about trying to understand what was happening. Drawing on in-depth interviews with household heads and community leaders, and focus groups with community members, this article describes the reasons underpinning the reticence to take up residence in the Star Homes and outlines their implications for other studies of improved housing in rural Africa and beyond.

## Methods and setting

### Study location

The population of Mtwara Region, in southeastern Tanzania, close to the border with Mozambique is around a million residents, mainly from the Makonde, Yao and Makua groups. The study communities are spread across the region and were selected using a cluster randomization design where the village was a cluster. All eligible households in a village (Cluster) who were identified through household eligibility survey conducted in 2019 were given opportunity to participate in a lottery where two household had a chance of winning a house. The lottery was conducted at a village meeting using a transparent format whereby the names of the heads of all eligible households were printed and placed in sealed envelopes. The crowd chose a literate child to run the lottery. After thoroughly mixing the envelopes in a transparent bucket, the child picked two envelopes and read the names aloud, as the recipients of the Star Homes. The remaining contestants were kindly invited to participate as comparison houses. This was the first project of its kind in the region to attempt the large-scale construction of houses in rural villages and in many of the selected communities, few health-related research or development initiatives had been implemented previously.

### Star homes

The design of the intervention houses drew on the results of the Magoda pilot project mentioned earlier, including the preferences of residents [5] and was undertaken by the Danish architecture firm, Ingvartsen (http://ingvartsen.dk/) in consultation with clinicians, entomologists, public health advocates and social scientists. The critical elements of the house design are described in Fig 1. Openings in the façade are covered by shade net, an industrial polyethylene insect mesh typically used in agriculture, and reinforced by wire mesh backing fixed in aluminium frames. All openings are fixed, except one panel on the first floor that can be opened as a means of escape in the event of a fire.

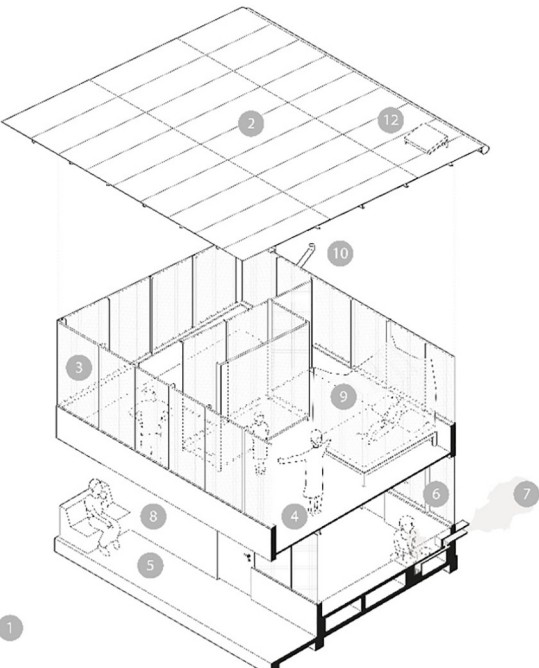
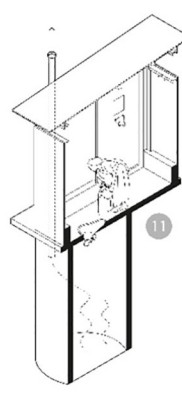

**Fig 1. Critical elements of the Star Home adapted from [9].** To reduce malaria, enteric and respiratory tract infections and to improve indoor climate and safety the star home includes the following critical structural components: 1) orientated to provide optimal shading throughout the day to keep the house cooler at night. (→ indoor climate). 2) lightweight and durable roof with partially closed eaves to reduce the entry of malaria vectors. (→ malaria, indoor climate). 3) facade and openings screened to reduce insect entry while assuring airflow. One panel is openable to facilitate egress in the event of a fire (→ malaria, diarrhoea diseases, indoor climate, safety). 4) use of lightweight, low thermal mass materials to reduce heat transmission and lower indoor temperatures at night to facilitate bed net use (→ malaria, indoor climate). 5) raised concrete ground floor which can be easily cleaned increases hygiene and reduces the risk of enteric and soil-transmitted infections. (→ diarrhoea diseases). 6) screened indoor cooking area with means to remove smoke to reduce indoor pollution which doubles as a screened seating area during the evening. (→ respiratory infections, malaria). 7) An improved cooking stove with chimney to reduce smoke inhalation indoors. (→ respiratory infections). 8) a protected lockable storage area to reduce rodent infestations and provide a feeling of security. (→ safety, diarrhoea diseases). 9) sleeping areas with bed nets raised to the first floor improving airflow, comfort while reducing mosquito density. Rooms used predominantly during the day are located on the ground floor therefore the bedrooms act as a buffer zone, reducing daytime temperature (→ malaria, indoor climate). 10) a water harvesting system, including a first flush mechanism, which allows the collection of rainwater from the roof, filtering, and covered storage. (→ diarrhoea diseases). 11) A ventilated outdoor fly-proof pit latrine with self-closing pan. (→ diarrhoea diseases). 12) solar power providing electric light at night and USB charging for mobile devices. (→ safety, health access).

After several prototype buildings had been evaluated by architects and engineers in an initial testing phase, construction was scaled up in selected villages, in Mtwara District, Mtwara. Target families resided in traditional mud walled, earth floor and thatched-roofed houses without electrification or piped water. Even though that was not intended the target families turned out to be the poorest members of communities. A total of 110 Star houses were built with 440 traditional homes used as a comparator group. Three children under 13 years were enrolled in the clinical surveillance in each house, sufficient to detect a statistically significant difference in malaria incidence between the two study groups over three years [9].

## Study design

This descriptive qualitative study drew on several data collection techniques including in-depth interviews and focus group discussions. A Tanzanian social scientist (JM) with over 10

years of experience of health-related research conducted interviews and focus groups with a range of respondents from across purposively selected study villages.

## Respondent recruitment

For the in-depth interviews, respondents were recruited from ten villages that had been identified by project research assistants (charged with conducting health surveillance and assisting with handover of the houses) as locations where there was wariness to move into the new houses or the use of houses was particularly inconsistent. Respondents were purposively sampled with the aim of interviewing household head of all Star Homes in the ten village. Further respondents, including village leaders who were involved in or aware of the project were also identified across the ten villages. Additional community members were invited to join focus groups held in the target villages. Participants were recruited based on their involvement in the project as a member of a comparison household (selected to participate based on the same eligibility criteria as the Star Homes but not "winners" in the lottery) and household heads of neighboring houses of the Star Homes. Respondents were recruited with the assistance of the research assistants and the District Malaria Focal Person (A coordinator of malaria transmission prevention and control in the district). The recruitment was done between 1$^{st}$ and 10$^{th}$ of October 2021 and interviews were conducted between 11$^{th}$ - 29$^{th}$ October 2021.

## Data collection techniques

In-depth interviews (Star Homes household heads and community leaders) and focus group discussions (comparison household heads and community members) were conducted using topic guides. The guides were based on those used in the Tanga pilot study but heavily adapted to examine the reasons for homeowners' reticence to enter the Star Homes and the wider community responses to the houses and the project. All interviews and focus group discussions were conducted by the author in confidence. The participants used codes throughout the interviews, although there was still a possibility of the author to identify some of the participants during and after data collection.

## Data processing and analysis

All interviews and focus groups were conducted in Swahili or Makonde and were audio-recorded. Recordings were transcribed and translated verbatim to English for qualitative content analysis. A codebook was developed based on the initial research questions of the ancillary qualitative research (which focused on the acceptability and use of the houses) and adapted as topics emerged during data collection, particularly regarding exploring the stories about 'Freemasons' and how they affected decisions to occupy the houses. All transcripts were read several times and coded line-by-line using QSR NVivo.

## Ethical approval

The Star Homes study protocol and the ancillary qualitative research has been approved by the National Research Ethics Committee of Tanzania on 17 June 2021 (reference NIMR/HQ/R.8a/Vol.IX/3695) and the Oxford Tropical Research Ethics Committee on 2 July 2020 (reference #533–20. All respondents provided written informed consent to participate in interviews or focus group discussions.

## Results

In-depth interviews were conducted with nine heads of the Star Homes and one adult member of the 10[th] Star Homes (because the household head was unavailable) in the ten selected communities. In these communities, a further six interviews were conducted with community leaders, selected based on their knowledge of and participation in the Star Homes project. Furthermore, 30 community members participated in four focus group discussions (Table 1), with an average duration of one and half hours. In these four villages, the focus groups included household heads of comparison houses (involved in the project but not recipients of a Star Home) and other community members. Of all respondents, around one third were female (14/46).

Several themes emerged as relevant to decisions about occupying the Star Houses. Reports of 'Freemasons' were particularly prominent and linked to other concerns. The impacts of the freemasonry accusations were dynamic and their impact on decision-making complicated by the fact that decisions were taken within household relationships, in communities where the stories were circulating and in settings where project staff were providing information and seeking to address the accusations. There were other reported practical reasons for delays in occupying the Star Homes: a custom of only moving in as a couple, a lack of furniture for the new home, complaints about the indoor temperature in the new home during June/July, the coldest months of the year in Mtwara, relocation to the *shamba* (family farm) or elsewhere.

### Project staff as 'Freemasons' and other accusations

Respondents described how other community members labelled Star Homes study staff as 'Freemasons' and how the residents of Star Homes were placing themselves at risk from 'Freemasons" nefarious activities. The potential harm was explained in general terms of risk of 'something bad', death or 'Freemasons' demanding 'blood sacrifice', particularly from the household's children. Although respondents reported personal concerns about the veracity of these stories, the details they conveyed were usually attributed to other community members. Respondents' own descriptions of the 'Freemasons' or their activities were rather vague. A couple of respondents referred to 'Freemasons' as wealthy outsiders who give support to communities, only for that support to be repaid through blood sacrifice. The involvement of 'Freemasons' was hence explained in terms of a way to understand why a house would be given for free: something must be given in return.

> They say that these 'freemason' people need blood. . .I should propose one child, sister or other relative every year to be sacrificed with their blood sucked until they die.

Interview with a female head of a Star Home household

(45–50 years old)

**Table 1. In-depth interview and focus group participants.**

| Data collection | n | Category of participants | | | n |
|---|---|---|---|---|---|
| | | | Females | Males | Total |
| IDIs | 10 | Star homes household heads/member | 4 | 6 | 10 |
| | 6 | Community leaders | 1 | 5 | 6 |
| FGDs | 4 | Comparison household heads | 5 | 8 | 13 |
| | | Community members/neighbours | 4 | 13 | 17 |
| **Total** | **20** | | **14** | **32** | **46** |

IDI = in depth interviews; FGD = focus group discussion

Further concerns about the houses were linked to related stories circulating in the communities. One was about the supposed presence of a secrete room in the houses, which was described as a place for the 'Freemasons' to sleep, where a blood-sucking 'jinx' resided, or as a dangerous place into which residents would disappear. Rather than the 'Freemason' label, sometimes community members reportedly called project staff 'Chitwadi' or 'Vitwadi'. These terms were for supernatural beings: Chitwadi were said to take souls and Vitwadi take away children's blood and transfer it abroad. There was also mention of another housing project elsewhere in which residents had been provided houses and witchcraft was involved.

*. . .there is a room, which is not opened. . . they keep a jinx there. . .it will suck your blood. If it sucks your blood, you will just fall down and die.*

Interview with a male Star Homes household head

(35–40 years old)

## 'Street words'

Suspicions regarding the project and project staff were said to come from other community members and respondents explained the accusations in terms of talk on the 'streets' or 'street words'. A couple of respondents referred to 'youth' spreading the stories, particularly when smoking and drinking in groups. Talk about 'Freemasons' was sometimes labelled as rumors and said to circulate in the market and discussed when collecting firewood or water. Those who talked about 'Freemasons' were depicted as not being 'enlightened' or lacking knowledge or information about the project. They were hence placed in contrast to those who had experience of or understood the project. Community leaders saw part of their role to provide information about the project to counter the 'Freemason' accusations. One such leader also described frustration at not having the necessary information to argue against the stories he heard in the community. Many respondents were dismissive of the accusations and described joining the project despite them.

*The families for whom the houses were constructed, their fear was built by the community. . .when they met at the water wells, the only story was the houses with freemason rumors. . .when collecting firewood, the only story is the same. . .in the market, the same story.*

Female focus group participant

(65–70 years old)

## The impact of the accusations

Community members' lack of exposure to other research in the area was viewed as contributing to the 'Freemason' accusations. The possibility that outsiders would undertake a study, which entailed providing benefits, such as a house, was unheard of and prompted suspicions about ulterior motives. Such accusations persisted even though the Star Homes were allocated by lottery and respondents reported being satisfied with this method of random allocation. There was also the suggestion that the accusations were driven by envy.

*I: What are village things? I don't know. . .*

*R: You know village things, when you hear somebody got something, another person may have something personal with them. . .so, you see, so in order to implant the idea and the other one stays afraid and to reject it*

*I: Why do they hate [the other person]?*

*R: Maybe he also wants the [Star Home], but it is a lottery.*

Interview with a female Star Home household head

(40–44 years old)

The impact of these stories on occupying the Star Home was rather fluid. Respondents were often vague about whether they were convinced by them. Suspicions regarding nefarious consequences following moving into the new houses were usually defused in discussions with project staff.

*I: Okay. Explain to me: how did you obtain the house?*

*R: The first time, I refused. They told me that [project staff] are freemasons*

*I: Who told you that?*

*R: It is just news in the streets*

*I: Okay*

*R: But when they arrived, and they informed me, I accepted after they cleared my worries. They told me they are not freemasons*

Interview with a female head of a Star Home household

(40–44 years old)

In one household, concerns about the project, fueled by the 'Freemason' accusations, remained after moving in and they therefore hid the children to avoid the weekly health surveillance. In another case, following the successful relocation into the Star Home, there were complaints of supernatural forces and the daughter of one family reported that nefarious forces dragged her into the latrine pit. In this instance, the household head sought assistance from the project to pay for a ceremony to rid the house of the malevolent forces.

Disagreements about the 'Freemason' accusations and their implications for living in a Star Home had life-changing implications for some project participants. For example, a couple divorced after the husband moved into the new with their children house despite his wife's protests. At least one other couple divorced over the decision whether to move into the Star Home. In this case, the husband found himself isolated by his neighbors who shared the fears of his ex-wife regarding the involvement of 'Freemasons' in the construction of the new house. The husband reported that over time the neighbors appeared to have changed their minds, accepted the new building, and even appreciated some of the advantages of the Star Home. Besides threats from neighbours, some Star Home residents had to contend with isolation, including from their extended family.

*At the beginning [my wife] was reluctant. I was ready to leave her and the children; I told her that I am not educated but I do understand [about the project] and I will not put my family at risk so as I can get something.*

Interview with a male household head

(44 years old)

## Discussion

The interviews and focus groups highlighted how, in communities around the Star Homes, project staff were widely associated with 'Freemasons' during the construction and handover period. These connections were generally not made by the respondents but rather the accusations attributed to other community members. The respondents focused on a knowledge or experience deficit among those who they blamed for making the accusations but also described them as envious and hoping to discourage people from accepting the Star Home so that the house may be re-allocated. The stories were also embedded in assumptions of reciprocity and suspicions about the motives of the project, explained in terms of a lack of local experience of other studies. The impact on decisions about occupying the Star Homes was often discussed indirectly, referring to others' concerns or reticence. When it was addressed directly, it was part of a dynamic process with the influence of project staff or relatives playing a role. Hence the relationship between the presence of 'Freemason' accusations and the reticence to occupy the houses was never straightforward. The stakes for the recipients of Star Homes were also high. By accepting the houses, they ran the risk of physical harm (from the 'Freemasons'), damage to their relationships, and social isolation because they became associated with the 'Freemasons' or because of envy in the wider community.

The reticence of houseowners to occupy the novel design homes was unexpected by the Star Homes project staff. Based on the experience during the pilot study in Magoda, the research team anticipated that the new homes would be well received [5]. The situation in Magoda was however different in terms of the relationship between project staff and the community before the first house was built. One team member (JK) and his family had visited Magoda for over two decades as part of several projects. He, plus other team members, had been involved in renovating the local school, improving teachers' housing and had been instrumental in constructing a community dispensary. In Magoda, after long-term involvement of project staff, there was no assumption about reciprocity in exchange for the houses. By contrast, the researchers had never visited Mtwara before the study start. Furthermore, the communities in Mtwara had had comparatively little exposure to medical or other researches.

When the project conducted a lottery for households who would receive a Star Home, only community members living in traditional homes–mud-walled dwellings with earth floors and thatched roofs–were included. These households were from the lowest socio-economic strata of the communities [10]. Only households with disposable income are generally able to change their homes with corrugated iron roofs, brick walls, concrete floors, for example. Hence, receiving one of the Star Homes represented a considerable improvement in the living circumstances and a significant change to the social and economic order of the community. Such an upheaval understandably led to questions from the households who received a home and other community members, particularly given the relatively lack of local experience with research. The motives of the project staff were queried, with such skepticism summed up in the Makonde saying, '*Cha bure cha kudoroja*' meaning 'what you get for free has its consequences'. That only the poorest received households and the random nature of the allocation likely contributed to the envy that respondents described among other community members.

The term 'Freemason' is used to denote members of a 'secret society', which is paradoxically well-known throughout Tanzania and has become a catch-all label for nefarious, supernatural

forces. Strangers with nefarious motives have not only been described in Tanzania but rather are a recurrent motif in mythology and popular culture [11]. There is extensive literature on the ever-present reality of supernatural entities with bad intentions in the consciousness of rural (and urban) communities in Africa [12, 13]. This role has been historically played by devils and witches of both sexes. In Tanzania, the idea of the witch or devil has evolved with transition from the command economy in the socialist period of Ujamaa (familyhood) to the 'magic' of the free market in the post-Nyerere neo-liberal economy [14].

Following the transition to a free market economy in Tanzania, the success of businessmen (male or female) has been best explained by '*chuma ulete*', a Swahili term meaning 'stealing money with the help of witchcraft' [14]. It is a given that wealth comes at the expense of the poor and supernatural forces are evoked to make sense of who wins and who loses. Historically, successful farmers were suspected (and punished) for syphoning off the grain of their less fortunate colleagues in supernatural ways [13]. 'Freemasons', a secret society aiming to help their brethren to achieve success in business, have become a preferred target in rationalizing the inexplicable accumulation of wealth and business success. Hence, viewing the project team as freemasons likely helped some community members make sense of why some households received Star Homes–gaining wealth and standing–and others did not.

The socio-cultural anthropologist Harry G. West has written about sorcery among Muedans, Makonde people living close to Mtwara in the north of Mozambique [15]. West described an agricultural credit scheme which allowed economically advantaged Muedans to acquire tractors, trucks, and grain mills below market cost. It was suggested that new owners had gained and sustained these goods with the help of witches to whom they had sacrificed members of their own family–often children. Within a short time, trucks, tractors, and grain mills started to break down. The owners reported being told that witches had paralyzed the machines by 'stuffing human skulls and body parts inside them'. According to West, the Makonde not only accused others of perpetrating witchcraft but simultaneously undid them by declaring knowledge of what they were up to [15].

To address the reluctance to occupy the houses and the accusations of freemasonry, the study team intensified its community engagement activities. The remedial action undertaken by the research team consisted of suspending research activities for six months to allow research assistants to build trust with study households and more broadly the study villages. This programme of intensive engagement in the community involved three main elements: sporting events; research assistants participating in everyday life; and a radio drama programme. The sports events in the communities pitted project staff again community members in football and netball matches. These games provided indications that no one in the project team gets help from supernatural forces: would a football team with access to superpowers lose two games, and end three games with a draw? Research assistants also took time to spend with the families; visiting for a day, helping to prepare food and spending time with the children. With the assistance of a specialist in community drama, the project team designed a series of radio plays to spread awareness about the aims of the project. Initially planned as community performances, COVID-19-related concerns led the team to broadcast the plays over the popular local radio stations. Community members acted out the script and there were opportunities for listeners to phone in and leave question or comments.

Researchers involved in clinical trials and other types of studies in Africa and beyond often receive accusations that–certainly for an outsider–are at odds with the usually benevolent nature of their work. These accusations take various, sometimes overlapping forms: conducting sterilization campaigns; stealing blood; satanism; being 'Freemasons' etc. [16–20]. It is not uncommon for the stories to have an impact on study recruitment and retention, and the staff usually respond by increasing engagement with the target communities, providing more

information and seeking to build trust [17]. There is however no single reliable formula for how to engage communities and address such accusations. Different communities require different approaches [21, 22].

What is common to many RCTs is that–by their very nature–they create inequalities. Whether through applying recruitment criteria that, for example, exclude children based on an age limit, or through a random lottery, there are those who benefit and those who do not [23]. In the case of a project that tests improved housing and targets the poorest in communities: the difference between winning and losing is not only pronounced for individual but it has significant implications for local social and economic hierarchies. Perhaps it should not have come as a surprise–given the propensity of Makonde (and other) communities to draw on supernatural narratives to make sense of inequalities–the project staff would be subject to these accusations. This is a key lesson for other projects that involve housing interventions in poor communities: the stakes are high, and researchers must be prepared for potential divisions and the implications they have.

## Strengths and limitations

The study was strengthened by the inclusion of several respondent types, including residents and non-residents of Star Homes, across ten communities in Mtwara Region. Furthermore, interviews and focus groups were conducted within a few weeks of project staff identifying the lower-than-expected occupancy of the houses. Thus, their recollections of the 'Freemasons' accusations and decisions about occupancy were likely fresh. The study did not aim to quantitatively assess the occupancy of the Star Homes and, as has been described, residency status was difficult to classify in a binary manor: some residents were either inconsistent over time or in terms of the household members present. Nonetheless, the study is limited by the absence of detailed description of the process of individual households entering the homes as they did so.

## Conclusion

In Mtwara Region, recipients of Star Homes and members of the communities in which they were built described how, during the handover period and beyond, project staff were accused of being 'Freemasons', members of a group with perceived nefarious motives. These accusations–usually attributed to third parties–were embedded in assumptions of reciprocity and explained in terms of knowledge deficit, envy, and a lack of familiarity with research in the region. The accusations' impact on decisions about occupying Star Homes was far from straightforward, with decisions influenced by interactions with project staff and relatives. The stakes for Star Homes residents were high particularly because they were among the poorest in the communities. The results contrast with previous experience of a housing intervention in Magoda (northeastern Tanzania) and indicate that long-term engagement with communities can ease concerns and suspicions. Large-scale housing projects, such as Star Homes in Mtwara Region can take a proactive approach to engagement, which focuses on building relationships and providing information through recognizable voices and formats. Given the stakes at play in studies that involving housing interventions, research teams should be prepared for the social upheaval such their interventions can cause.

## Supporting information

**S1 File. Inclusivity in global research questionnaire.**
(DOCX)

## Acknowledgments

The authors would like to thank the respondents and the communities where the Star Homes project was undertaken. The authors would also like to acknowledge the contribution of the CSK staff who transcribed verbatim and translated to English the interview and focus group audio recordings.

## Author Contributions

**Data curation:** Judith Meta, Christopher Pell.

**Formal analysis:** Judith Meta, Christopher Pell.

**Funding acquisition:** Lorenz von Seidlein.

**Investigation:** Judith Meta, Salum Mshamu, Lorenz von Seidlein, Christopher Pell.

**Project administration:** Salum Mshamu.

**Supervision:** Salum Mshamu, Christopher Pell.

**Validation:** Christopher Pell.

**Writing – original draft:** Judith Meta, Christopher Pell.

**Writing – review & editing:** Salum Mshamu, Salma Halifa, Arnold Mmbando, Hannah Sloan Wood, Otis Sloan Wood, Thomas Chevalier Bøjstrup, Nicholas P. J. Day, Jakob Knudsen, Steven W. Lindsay, Jacqueline Deen, Lorenz von Seidlein, Christopher Pell.

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
