## [Decision Letter · Decision Letter 0]

3 Jul 2023

PGPH-D-23-00654

Understanding reticence to occupy free, novel-design homes: a qualitative study in Mtwara, Southeast Tanzania

Dear Dr. Mshamu,

Thank you for submitting your manuscript to PLOS Global Public Health. After careful consideration, we feel that it has merit but does not fully meet PLOS Global Public Health’s publication criteria as it currently stands. Therefore, we invite you to submit a revised version of the manuscript that addresses the points raised during the review process.

EDITOR'S Comments:

Please address the suggested revisions from both reviewers to add clarity to your methods, reduce repetition between your results and discussion, and increase the interpretation and application of your findings.  

You should also take this opportunity to review your article's adherence to the data availability policy and reporting guidelines for qualitative studies as stated in the PLOS submission guidelines.  https://journals.plos.org/globalpublichealth/s/submission-guidelines#loc-qualitative-research If you are not able to make the underlying data for this study available, please justify this to the editorial office and propose an alternative plan https://journals.plos.org/globalpublichealth/s/data-availability#loc-acceptable-data-access-restrictions. 

We look forward to receiving your revised manuscript.

Kind regards,

Sarah E. Brewer, PhD

Academic Editor

Journal Requirements:

1. Please include a complete copy of PLOS’ questionnaire on inclusivity in global research in your revised manuscript. Our policy for research in this area aims to improve transparency in the reporting of research performed outside of researchers’ own country or community. The policy applies to researchers who have travelled to a different country to conduct research, research with Indigenous populations or their lands, and research on cultural artefacts. The questionnaire can also be requested at the journal’s discretion for any other submissions, even if these conditions are not met.  Please find more information on the policy and a link to download a blank copy of the questionnaire here: https://journals.plos.org/globalpublichealth/s/best-practices-in-research-reporting. Please upload a completed version of your questionnaire as Supporting Information when you resubmit your manuscript.”

2. Please send a completed 'Competing Interests' statement, including any COIs declared by your co-authors. If you have no competing interests to declare, please state "The authors have declared that no competing interests exist". Otherwise please declare all competing interests beginning with the statement "I have read the journal's policy and the authors of this manuscript have the following competing interests:"

3. Please provide a/amend your detailed Financial Disclosure statement. This is published with the article. It must therefore be completed in full sentences and contain the exact wording you wish to be published.

a. State the initials, alongside each funding source, of each author to receive each grant. For example: ""This work was supported by the National Institutes of Health (####### to AM; ###### to CJ) and the National Science Foundation (###### to AM).""

4. In the online submission form, you indicated that "All data presented in this article will be accessible whenever required. The transcripts and quotations of this qualitative work can be shared as required.". 

Additional Editor Comments (if provided):

Reviewers' comments:

Reviewer's Responses to Questions

**Comments to the Author**

1. Does this manuscript meet PLOS Global Public Health’s publication criteria? Is the manuscript technically sound, and do the data support the conclusions? The manuscript must describe methodologically and ethically rigorous research with conclusions that are appropriately drawn based on the data presented.

Reviewer #1: Yes

Reviewer #2: Yes

2. Has the statistical analysis been performed appropriately and rigorously?

Reviewer #1: N/A

Reviewer #2: N/A

3. Have the authors made all data underlying the findings in their manuscript fully available (please refer to the Data Availability Statement at the start of the manuscript PDF file)?

Reviewer #1: No

Reviewer #2: No

4. Is the manuscript presented in an intelligible fashion and written in standard English?

Reviewer #1: Yes

Reviewer #2: Yes

5. Review Comments to the Author

Reviewer #1: The findings of the study are very intriguing. This study highlights the continued prevalence of superstitions in underprivileged and poor communities. The ‘results’ section is well written. However, there are a few points that I would like to raise

1. A few studies should be mentioned in the third paragraph of the ‘background’ section that previously investigated the existing superstitious concepts like ‘freemansons’ and supernatural forces in the poor and uneducated communities in Tanzania or Africa.

2. The interview guide of the in-depth interview should be discussed lucidly.

3. The detailed process of the focus group discussions (FGDs) is not mentioned. The authors should mention how the FGDs were conducted, what questions were discussed with the respondents.

4. The findings correspond to the objectives of the study. However, it appears that interpretations are written repeatedly in the paragraphs no 2-6 of the ‘discussion’ section. I would like to see it rewritten in a crisp manner not repeating certain things.

5. Nowhere in the discussion section I see the mention of role of low or little education behind the reluctance and fear of the residents in Mtwara to occupy the free, novel-design houses. It should be interpreted.

Reviewer #2: This manuscript presents and interprets qualitative findings from a housing intervention study, where the study experienced challenges in acceptance and use of the newly constructed houses by community members.

I found the paper to be well written, and appreciate the authors' openness to share the challenges experienced during this trial, so that their experiences may inform and guide other researchers.

I have only a couple of minor suggestions:

1. In line 155 an eligibility survey is mentioned, and in line 138 it is stated that the target families were the poorest members of the community, even though that was not the intention. Could you unpack this a little, perhaps stating in line 155 the specific eligibility criteria for a household to be entered into a lottery? Were the vast majority of households in each village considered eligible, or only a small subset? This may also help to contextualise the later statements about envy of those who received the new homes

2. Line 168. Suggest being consistent with the pilot study location name - here it is Tanga, previously it was Magoda.

3. Line 298. This sentence requires rewording

6. PLOS authors have the option to publish the peer review history of their article (what does this mean?). If published, this will include your full peer review and any attached files.

**Do you want your identity to be public for this peer review?** For information about this choice, including consent withdrawal, please see our Privacy Policy.

Reviewer #1: No

Reviewer #2: No

---

## [Decision Letter · Decision Letter 1]

25 Sep 2023

Understanding reticence to occupy free, novel-design homes: a qualitative study in Mtwara, Southeast Tanzania

PGPH-D-23-00654R1

Dear Mr. Mshamu,

We are pleased to inform you that your manuscript 'Understanding reticence to occupy free, novel-design homes: a qualitative study in Mtwara, Southeast Tanzania' has been provisionally accepted for publication in PLOS Global Public Health.

Best regards,

Sarah E. Brewer, PhD

Academic Editor

Reviewer Comments (if any, and for reference):

Reviewer's Responses to Questions

**Comments to the Author**

1. If the authors have adequately addressed your comments raised in a previous round of review and you feel that this manuscript is now acceptable for publication, you may indicate that here to bypass the “Comments to the Author” section, enter your conflict of interest statement in the “Confidential to Editor” section, and submit your "Accept" recommendation.

Reviewer #1: All comments have been addressed

2. Does this manuscript meet PLOS Global Public Health’s publication criteria? Is the manuscript technically sound, and do the data support the conclusions? The manuscript must describe methodologically and ethically rigorous research with conclusions that are appropriately drawn based on the data presented.

Reviewer #1: Yes

3. Has the statistical analysis been performed appropriately and rigorously?

Reviewer #1: N/A

4. Have the authors made all data underlying the findings in their manuscript fully available (please refer to the Data Availability Statement at the start of the manuscript PDF file)?

Reviewer #1: Yes

5. Is the manuscript presented in an intelligible fashion and written in standard English?

Reviewer #1: Yes

6. Review Comments to the Author

Reviewer #1: The authors have addressed all the comments carefully and amended the manuscript accordingly. In my opinion, this study now has the potential to be accepted for publication.

7. PLOS authors have the option to publish the peer review history of their article (what does this mean?). If published, this will include your full peer review and any attached files.

**Do you want your identity to be public for this peer review?** For information about this choice, including consent withdrawal, please see our Privacy Policy.

Reviewer #1: No
